# Exploring the Language Used to Describe Older Patients at Multidisciplinary Cancer Conferences

**DOI:** 10.3390/cancers16081477

**Published:** 2024-04-12

**Authors:** Valerie S. Kim, Anthony Carrozzi, Efthymios Papadopoulos, Isabel Tejero, Thirisangi Thiruparanathan, Nathan Perlis, Andrew J. Hope, Raymond W. Jang, Shabbir M. H. Alibhai

**Affiliations:** 1Temerty Faculty of Medicine, University of Toronto, Toronto, ON M5S 1A8, Canada; valerie.kim@mail.utoronto.ca (V.S.K.); anthony.carrozzi@mail.utoronto.ca (A.C.); 2Department of Medicine, University Health Network, Toronto, ON M5G 2C4, Canada; 3School of Kinesiology, Louisiana State University, Baton Rouge, LA 70802, USA; epap@lsu.edu; 4Department of Geriatrics, Hospital del Mar, 08003 Barcelona, Spain; itejero@psmar.cat; 5Department of Nursing, University Health Network, Toronto, ON M5G 2C4, Canada; thirisangi.thiruparanathan@mail.utoronto.ca; 6Division of Urology, Department of Surgical Oncology, University Health Network, Toronto, ON M5G 2C4, Canada; nathan.perlis@uhn.ca; 7Department of Radiation Oncology, University of Toronto, Toronto, ON M5T 1P5, Canada; andrew.hope@uhn.ca; 8Radiation Medicine Program, Princess Margaret Cancer Centre, Toronto, ON M5G 2M9, Canada; 9Division of Medical Oncology and Hematology, Princess Margaret Cancer Centre, Toronto, ON M5G 2M9, Canada; raymond.jang@uhn.ca; 10Department of Medicine, University of Toronto, Toronto, ON M5G 2C4, Canada; 11Institute of Health Policy, Management, and Evaluation, University of Toronto, Toronto, ON M5T 3M6, Canada

**Keywords:** multidisciplinary cancer conferences, tumour boards, geriatric oncology, geriatric assessment, frailty, communicate, language

## Abstract

**Simple Summary:**

Older adults with cancer are often the subject of much discussion at multidisciplinary cancer conferences (MCCs), yet little is known about the language used to describe frailty and other geriatric considerations at these meetings. Our objective was to explore how MCC presentations depict older patients. We found that MCCs frequently referred to comorbidity burden and projected treatment tolerance on the basis of subjective evaluations, rather than comprehensive geriatric assessments. We also noted that mentions of surrogate measures for frailty, such as chronological age and performance status, varied between tumour sites, presenter specialties, and presenter training levels. Overall, our results suggest that MCCs predominantly rely on age-based descriptions and, thus, engender risk of age-biased decision making. This work may guide future efforts aimed at standardizing the language at MCCs to include more objective terms and validated tools for considering different geriatric domains when discussing older adults with cancer.

**Abstract:**

Older adults with cancer often present with distinct complexities that complicate their care, yet the language used to discuss their management at multidisciplinary cancer conferences (MCCs) remains poorly understood. A mixed methods study was conducted at a tertiary cancer centre in Toronto, Canada, where MCCs spanning five tumour sites were attended over six months. For presentations pertaining to a patient aged 75 or older, a standardized data collection form was used to record their demographic, cancer-related, and non-cancer-related information, as well as the presenter’s specialty and training level. Descriptive statistics and thematic analysis were employed to explore MCC depictions of older patients (*n* = 75). Frailty status was explicitly mentioned in 20.0% of presentations, but discussions more frequently referenced comorbidity burden (50.7%), age (33.3%), and projected treatment tolerance (30.7%) as surrogate measures. None of the presentations included mentions of formal geriatric assessment (GA) or validated frailty tools; instead, presenters tended to feature select GA domains and subjective descriptions of appearance (“looks to be fit”) or overall health (“relatively healthy”). In general, MCCs appeared to rely on age-focused language that may perpetuate ageism. Further work is needed to investigate how frailty and geriatric considerations can be objectively incorporated into discussions in geriatric oncology.

## 1. Introduction

Older adults comprise a growing majority of patients with cancer, yet they remain disproportionately underrepresented in research trials and evidence-based guidelines [1,2]. As a result, clinical decision making for this demographic can be complicated and nuanced, particularly given the heterogeneity in their medical comorbidities, cognitive abilities, and functional levels [3,4,5]. 

Against this backdrop, older adults are often the subject of much discussion at multidisciplinary cancer conferences (MCCs) or “tumour boards”, where health professionals with complementary areas of expertise convene to deliberate cases, review guidelines, and coordinate treatment plans [6]. According to a recent umbrella review, this forum can facilitate guideline-concordant diagnosis and treatment [7]. By fostering communication between cancer care team members and promoting discourse on best practices, MCCs represent key opportunities for collaborative decision making and high-quality care [6,7]. 

At these meetings, a high-level summary about each patient is presented and discussed in a time-sensitive manner [6,7]. The details shared and the language used can carry significant connotations and implications for treatment decisions: depictions of patients as “elderly” or “old”, for instance, can influence clinicians’ perceptions and subsequently their recommendations [6,8]. In fact, patients who are described as older may be less likely to be referred for adjuvant chemotherapy [9]. According to a recent literature-based perspective piece, age-biased decision making and stereotyping appear to pervade conversations related to diagnostic and therapeutic recommendations for older adults with cancer [10,11,12]. In one report, a reliance on disease-centred, rather than patient-centred, terminology when caring for this demographic emerged as one of the five common patterns of stigmatizing language [12]. The prevalence of such trends in the discussion of older adults and their care raises concerns about potential ageism [12]. 

In our clinical experience, most MCCs without the presence of a geriatrician or a geriatric oncologist tend to focus on oncologic factors with little mention of geriatric considerations [13]. For instance, mentions of activities of daily living (ADLs) remain lacking at these meetings, despite the relevance of this detail, not only to patients’ suitability for treatment, but also to their decision-making priorities [13]. While their comorbidities are generally reported in comprehensive lists using precise medical terminology, non-cancer-related characteristics are poorly described with vague phrases (for instance, “pretty fit”) [8,13]. In fact, the existing literature reveals that MCCs often allude to patients’ fitness as an indicator of their general physical state or their eligibility for a specific intervention, yet consensus on the definition of “fit” remains unclear among the different clinicians and specialties [8]. Although clinical guidelines have recommended the use of the geriatric assessment (GA) in evaluating frailty, it is not commonly implemented, and it remains unclear whether GA domains are even being discussed at MCCs, both for patients in general and for the select few who have undergone a formal GA [2,13]. 

To date, little research has explored how MCCs depict older patients with attention to their frailty levels and geriatric considerations [8,13,14,15,16,17,18,19]. This has left a major gap in our understanding of the impact of language on cancer care. For these reasons, our study aimed to explore how MCCs presented older patients, their frailty considerations, and their GA domains. 

## 2. Materials and Methods

### 2.1. Objectives 

The study’s primary objective was to explore whether and how frailty status and GA domains are discussed in relation to treatment decision making for older adults with cancer at MCCs. We sought to examine the depiction of frailty, in the form of explicit descriptions of patients as “frail” or “fit”, mentions of chronological age as a proxy for frailty status, and references to other synonyms for frailty or fitness. We additionally sought to assess the incorporation of the GA, specifically with the inclusion of two or more GA domains, into MCC discussions. 

As an exploratory objective, we also set out to compare case presentations by MCC site, by presenter specialty, and by presenter training level. The latter was included to potentially inform educational initiatives that may arise from our findings. 

### 2.2. Study Design and Setting

For this mixed methods study, we attended weekly virtual general gastrointestinal, genitourinary, head and neck, thoracic, and upper gastrointestinal MCCs at the Princess Margaret Cancer Centre in Toronto, Canada. These five MCC streams were selected after reviewing the number of annual referrals and identifying the tumour sites with the highest proportion of new patients aged 75 or older. This cutoff was selected based on the consensus among geriatric oncology experts and among study investigators that all patients aged 75 or older should undergo a GA [20]. 

At each MCC, a presiding chairperson oversees presentations, ensuring that relevant disciplines (e.g., radiology, pathology, medical/radiation/surgical oncology) are in attendance. Cases are presented by the physician (either the attending or a senior trainee, such as a resident or fellow) who submitted the case to MCC: they share pertinent clinical context and the reason for presentation. As relevant, imaging and pathology are reviewed by the respective disciplines. Subsequently, all attendees are invited to weigh in and offer input on the diagnostic or therapeutic question at hand. The chairperson facilitates these discussions and ultimately summarizes recommendations for each case.

For all sites, a general summary about the project was circulated a week in advance to MCC chairs and attendees, who were provided with the opportunity to opt out should they wish to be excluded from our data collection and analysis. 

Approval from the Research Ethics Board of University Health Network was obtained prior to study commencement. The requirement for informed patient consent was waived. 

### 2.3. Data Collection 

A data collection form (Appendix B) was developed and refined through iterative revisions to ensure comprehensive, standardized capture of data across the research team. As nonparticipating observers, alternating members of the team (V.S.K., medical student; A.C., medical student; E.P., postdoctoral researcher in exercise oncology; I.T., geriatric oncology fellow; and T.T., graduate nursing student) attended MCCs and electronically recorded our variables of interest for presentations pertaining to patients aged 75 or older. We organized these variables into two groups. The first group consisted of frailty-related variables: in the absence of a universally shared definition for frailty at these meetings, these variables included explicit terminology or assessment to describe patients as “frail” or “fit”, as well as synonyms for frailty status, including activity, mobility, appearance, overall health, treatment tolerance, independence, and performance status. The second group consisted of GA-related variables encompassing the following domains: functional status, physical performance, falls, comorbidities, medications, cognition, mood, social support, and nutritional status.

The data collection form included a checklist, where the research team could indicate whether a variable was mentioned, as well as an area for free text, where frailty and GA-related descriptions could be recorded verbatim. Definitions and examples for each variable are provided in Table 1. 

### 2.4. Data Analysis 

Once completed, data collection forms were analyzed using convergent parallel mixed methods. While the overall cohort included representation from five tumour sites, the site-, specialty-, and training level-specific analyses excluded the head and neck presentations given the small sample size (n = 4). For the remaining four tumour sites, we anonymized their identities, as we were primarily interested in potential differences between the sites, rather than specific details about the individual sites themselves. We then enumerated the MCC presentations that included mentions of frailty- and GA-related variables. We were specifically interested in the proportion of discussions that explicitly described patients as frail or fit, that used chronological age as a proxy for frailty status, that alluded to frailty with nonage synonyms (for instance, overall health, performance status, projected treatment tolerance), and that included mentions of two or more GA domains. For these outcomes of interest, we stratified our observations by MCC site, presenter specialty, and presenter training level and used chi-square tests (with the Fisher’s and Fisher-Freeman–Halton exact tests where applicable) to identify statistically significant differences. We also conducted a thematic analysis of terms and phrases shared during MCC discussions. One researcher (V.S.K.) reviewed all field notes to identify recurring themes in the representation of frailty descriptions and GA domains; synonyms for frailty and mentions of geriatric considerations were reviewed and consensually agreed upon by the core research team. 

## 3. Results

### 3.1. Presentations in Overall Cohort 

From January to June 2022, we attended 42 MCCs. A total of 75 case presentations pertained to patients aged 75 or older and included representation from the general gastrointestinal (n = 15), genitourinary (n = 18), head and neck (n = 4), thoracic (n = 19), and upper gastrointestinal (n = 19) sites. The mean age of included patients was 80.9 years (range 75–97 years), the majority (61.3%) were male, and more than half (55.5%) had a new diagnosis of cancer. With only a small proportion of cases (5.3%) centred around diagnostic uncertainty, the predominant reason (86.7%) for presentation was to deliberate options for treatment, which included chemotherapy (46.7%), radiotherapy (53.3%), surgery (49.3%), immunotherapy (5.3%), and best supportive care (8.0%). A small proportion of these discussions made mention of patient preferences (20.0%). Additional baseline demographic and clinical characteristics are reported in Table 2.

### 3.2. Descriptions of Frailty in Overall Cohort

Across the five tumour sites comprising our overall cohort, 20.0% of the 75 case presentations explicitly identified individuals as “frail” or “fit”. None, however, mentioned a validated assessment tool to explain how frailty status was determined. More frequently, discussions referred to patients’ age (33.3%), projected treatment tolerance (30.7%), and overall health (21.3%) to hint at suspected frailty (Table 3, Figure 1).

While age was consistently included as part of identifying data, it was often reiterated to suggest either fitness or frailty. For instance, one presenter described their patient as “an okay 86 [year-old]”, while another pointed out that their patient was “very fit … for her age”. Conversely, MCC discussions restated age to portray a patient as frail. In one instance, a patient was noted to be “87 [years-old], so elderly and frail”. 

Age was also highlighted to suggest poor treatment tolerance. A potential regimen was described as “a lot for an 82-year-old”, while some presenters directly stated that “chemotherapy [was] obviously not at play given his age” or that “at that age, we wouldn’t be doing [surgery]”. Other frailty-related descriptions tended to be vague, such as “not the most robust”, or “pretty healthy, very active”, and “looks good, moves well”. In most cases, little elaboration, if any, was provided to justify frailty assignments with objective measures of physical performance or functional independence, for instance. 

### 3.3. Mentions of GA Domains in Overall Cohort 

Across the 75 MCC discussions, 22 case presentations (31.0%) made reference to at least two GA domains. The most frequently mentioned GA domain was comorbidity status, which was included in more than half (52.0%) of the case presentations, predominantly to imply poor treatment candidacy. For example, one presenter argued that “surgery is not ideal because of [the patient’s] multiple comorbidities”. 

Beyond comorbidity burden, a medication list was provided for 21.3% of the cases. Social support (9.3%), functional status (6.7%), cognition (5.3%), nutritional status (4.0%), and mood (1.3%) were reported much less frequently. Falls risk was captured in only one MCC presentation, but none of the case discussions included objective measures of physical function (Table 3, Figure 2). 

Interestingly, one (1.3%) of the cases included mention of an assessment by geriatric oncology, specifically quoting their estimate of a patient’s five-year survival rate. A comprehensive GA, however, was not discussed during this MCC presentation or discussion. 

### 3.4. Case Presentations by MCC Site 

Across the four MCC sites included in this analysis, discussions often included both direct and indirect references to frailty status (Appendix A). From here on, the sites are labelled as A, B, C, and D, in no particular order. Varying proportions of the sites made explicit mention of frailty or fitness (26.7% of site A, 11.1% of site B, 26.3% of site C, and 15.8% of site D presentations; *p*-value 0.61). Frailty status was also indirectly implied by reiterating chronological age (53.3% of site A, 11.1% of site B, 36.8% of site C, and 36.8% of site D presentations; *p*-value 0.072) or by describing overall health (13.3% of site A, 11.1% of site B, 15.8% of site C, and 42.1% of site D presentations; *p*-value 0.11) and performance status (6.7% of site A, 11.1% of site B, 0% of site C, and 26.3% of site D presentations; *p*-value 0.069). Although none of the discussions included a validated prediction tool for estimating chemotherapy toxicity or surgical risk, MCC presenters and attendees frequently alluded to projected treatment tolerance across the tumour sites (40% of site A, 22.2% of site B, 15.8% of site C, and 47.4% of site D presentations; *p*-value 0.14). 

Inclusion of GA domains was highly inconsistent. Only comorbidity status was discussed at all sites, albeit with considerable variation (13.3% of site A, 94.4% of site B, 47.4% of site C, and 36.8% of site D presentations; *p*-value < 0.001). Across the sites, the frequency of more than two GA domains being discussed also varied substantially, with mentions in 13.3% of site A, 83.3% of site B, 5.3% of site C, and 21.2% of site D cases (*p*-value < 0.001). 

### 3.5. Case Presentations by Presenter Specialty 

Of the 71 presentations discussed in the general gastrointestinal, genitourinary, thoracic, and upper gastrointestinal meetings, 22 were led by medical oncologists, 7 by radiation oncologists, and 44 by surgical oncologists. The sum of presenters exceeds the total number of cases as some were jointly presented by multiple specialists. For cases that were jointly presented by physicians of different specialties (for instance, radiation oncology and surgical oncology), any mention of a variable of interest was counted toward each specialty’s total. Appendix A captures quantitative data from their presentations, stratified by specialty. 

Explicit descriptions of patients as frail or fit were included in 22.7%, 28.6%, and 15.9% of the presentations led by medical, radiation, and surgical oncologists, respectively (*p*-value 0.61). Across the three specialties, chronological age (36.4%, 42.9%, and 34.1%, respectively; *p*-value 0.94) and projected treatment tolerance (22.7%, 57.1%, and 34.1%, respectively; *p*-value 0.25) were mentioned more frequently to capture frailty status indirectly. A considerable number of presentations made mention of at least one GA domain (40.9% by medical, 57.1% by radiation, and 61.4% by surgical oncologists; *p*-value 0.30), but reference to more than two GA domains varied to a greater extent (22.7% by medical, 0% by radiation, 38.6% by surgical oncologists; *p*-value 0.085). A statistically significant difference in the presenters’ inclusion of a patient’s performance status was noted, with this variable being mentioned in 27.2% of the cases presented by medical and 4.5% of the cases presented by surgical oncologists, but none of the cases presented by radiation oncologists (*p*-value 0.019). 

### 3.6. Case Presentations by Presenter Training Level

Of the 71 presentations spanning the four tumour sites included in this analysis, 33 were presented by trainees and 40 by faculty members. As was the case for presenter specialty, the sum of presenters exceeds the total number of cases, as some were jointly presented by trainees and faculty. For this reason, any mention of a variable of interest was counted toward trainee’s and faculty’s totals. Appendix A includes the quantitative data from their presentations, stratified by training level. 

Direct mentions of frailty status were included in 21.2% and 17.5% of the presentations led by trainees and faculty members, respectively (*p*-value 0.77). Trainees and faculty members both offered indirect descriptions of frailty or fitness through chronological age (18.2% and 45.0%, respectively, *p*-value 0.024), overall health (21.2% and 22.5%, *p*-value 1.00), performance status (21.2% and 2.5%, *p*-value 0.019), and projected treatment tolerance (27.3% and 30.0%, *p*-value 1.00). Interestingly, the only statistically significant differences in mentions of frailty-related variables between training levels were for age and performance status, with the former being mentioned more frequently by faculty and the latter by trainees. 

With respect to GA domains, 54.5% of presentations by trainees compared to only 15.0% of presentations by faculty made reference to two or more domains (*p*-value < 0.001). Regardless of the presenter’s training level, comorbidity status was the most frequently mentioned domain.

## 4. Discussion

This study used a mixed-methods approach to explore the description of frailty considerations and GA domains in MCC case presentations pertaining to individuals aged 75 or older. Although a handful of patients were explicitly described as frail or fit, none of the discussions mentioned a validated assessment tool for identifying or measuring frailty (for instance, the Rockwood Clinical Frailty Scale, Geriatric 8, Vulnerable Elder Survey-13) [15,21]. Instead, presenters tended to focus on surrogate measures to hint at frailty, often with subjective terms based on individual clinicians’ perceptions of a patient’s appearance, projected treatment tolerance, or overall health. Rather than comprehensively covering different aspects of the GA, comorbidity status was heavily featured, with much less attention paid to the other domains. These trends held true across the tumour board sites, presenter specialties, and training levels. Statistically significant differences, however, were observed between MCC sites for the discussion of at least two GA domains, between presenter specialties for the discussion of performance status, and between presenter training levels for the discussion of age, performance status, and at least two GA domains. Overall, our findings suggest that MCC case presentations disproportionately rely on chronological age and comorbidity burden with a lack of standardized language to define and explore other geriatric considerations relevant to frailty. 

It is important to note, however, that chronological age has been previously studied and found to be negatively associated with poor clinical outcomes in oncology. The MACH-NC Collaborative Group, for instance, found that older patients, compared to their younger counterparts, had a statistically significant reduction in potential benefit from concomitant chemotherapy with loco-regional treatment for head and neck cancer [22]. Similarly, Coate et al. found worsening survival with increasing age in the setting of stage III non-small-cell lung cancer [23]. Yet, when performance status, comorbidity burden, and other confounding factors were controlled for in their multivariate model, no difference in survival was identified between age groups after aggressive treatment with curative intent [23]. This finding highlights the importance of not relying on chronological age alone, particularly as mentions of chronological age, depending on how it is discussed, may perpetuate ageism [24,25]. 

Despite growing recognition of the importance of language, particularly in the care of older adults, only a small body of work to date has delved into the actual words and phrases used in clinical decision making for this demographic [8,13,14,16,17,18,19]. At a tertiary hospital in Australia, Lane et al. found a predominance of disease-centred language during MCC case presentations on older adults, who were often depicted using nonobjective, potentially ambiguous descriptors (e.g., “fit”, “well”, and “good”); their comorbid medical conditions, in contrast, were discussed at greater length with precise medical terminology [8]. A prospective six-centre audit by Lakhanpal et al., also based in Australia, reported comorbidities as the most frequently mentioned geriatric domain (92%), while fewer references were made to other relevant variables, such as functional status with regard to ADLs (50%) and instrumental ADLs (26%) [13]. Two of the presented cases reached a decision to withhold treatment on the basis of advanced age alone [13]. Bolle et al. similarly found limited inclusion of patient-centred information, such as patient values and preferences, in MCCs conducted across five nonacademic hospitals in The Netherlands [14]. In that country, a single-site cross-sectional review by Festen et al. corroborated the poor representation of patient-centred information and the reliance on comorbidity-related information when depicting older individuals [18]. A single-centre study centred around a gynaecological cancer multidisciplinary team in the United Kingdom also noted that disease-specific details appeared to take precedence over patient-related factors, including patient choice [16]. The ensuing skewed discussions give rise to the possibility of age-biased decisions. In their observational study on MCCs at two French centres, Billon et al. concluded that nearly half of the therapeutic decisions for older adults with cancer were “age-adapted” (i.e., driven by age alone) [17]. Finally, Restivo et al. explored how the inclusion of nonmedical characteristics affected treatment decisions and practices at a French cancer centre [19]. They found that patients’ age and “likeability”, described using terms such as “nice” or “annoying”, were the most frequently mentioned variables; inclusion of these characteristics was positively associated with deferred decisions, suggesting that the addition of nonmedical information can serve to remind clinicians of the complexity inherent in caring for older adults [19]. 

These studies from around the world offer findings that are concordant with the results of our study. Despite the recommendation from the International Society of Geriatric Oncology to tailor treatment decisions to a patient’s health status as opposed to their chronological age, prior literature and our work suggest ongoing shortcomings in how older patients and relevant geriatric considerations—beyond age alone—are discussed [26]. The limited incorporation of the GA reflects another critical practice gap. While the use of the GA has been shown to increase the likelihood of cancer treatment completion, lower rates of hospitalization, improve patient-centred communication, and enhance patient satisfaction among older adults, our study reveals skewed, comorbidity-focused representation of the GA domains [27,28,29,30]. 

With our efforts to investigate this significant guideline-practice gap, our study possesses a number of strengths. First, this is the only study, to our knowledge, that compares the language at MCCs across various oncologic specialties and presenter training levels. This may reflect nuances in training practices or team cultures that contribute to heterogeneity in geriatric language use. It may also provide a glimmer of hope that some reporting trends are improving among trainees compared to faculty, since we observed significantly less reporting of age and more reporting of performance status among trainees. Second, this is the only paper to capture mentions of both frailty depictions and GA domains; while Lakhanpal et al. and Bolle et al. also examined the latter, neither examined how patients are described as fit or frail when discussing potential treatment plans [14,18]. Third, we included cases from various tumour sites to increase the generalizability of our results and to capture differences across sites. Fourth, our mixed methods approach enabled us to identify themes and quantify language use, providing a more fulsome picture of the words and phrases included at these meetings. Finally, this is the first study in North America to explore the depiction of older patients and their treatment considerations at MCCs [8,13,14,16,17,18,19]. 

Our findings must be interpreted with caution, however, in the context of our study’s limitations. First, the nature of our methodology left potential room for the observer effect, as the MCC presenters were informed in advance of our attendance. Consequently, our involvement, even without any active participation, may have altered how the healthcare providers communicated or behaved during MCC meetings. This may have led to optimistic estimates for our observed outcomes. Second, our data collection relied on field notes compiled by different research team members. To mitigate inter-rater differences, however, we had used a standardized data collection form, and we had undergone trials of joint data collection to ensure similar recording by all research team members. Third, our sample size was modest, with discrepancies between MCCs sites, notably with a small number of head and neck MCCs included due to scheduling challenges. Fourth, we focused exclusively on the discussions at MCCs without extending into post-meeting decisions or outcomes. As a result, the ultimate clinical impact of the language used at these MCCs remains unknown. 

There are several potential next steps with significant implications both in terms of clinical practice and research. For future clinical practice, our study highlights the need to mitigate the risk of ageism and improve how older patients are discussed beyond their chronological age. This demographic tended to be depicted using vague and subjective language, which begs the question of how clinicians can be encouraged to incorporate more objective descriptors in hopes of minimizing age-biased decision making [12]. For instance, one wonders whether a checklist comprising the GA domains would be helpful for MCC presenters to review for each geriatric case. This strategy may also help foster more comprehensive discussions that better address elder-relevant priorities, such as the preservation of functional autonomy and quality of life. Similarly, the presence and integration of a geriatrician at MCCs may offer richer conversations about geriatric considerations beyond cancer- or comorbidity-focused variables, although the feasibility of this approach may be dictated by local resources [14,31]. For future research, our study paves the way for further work with larger sample sizes that may reveal even more statistically significant differences between tumour sites, presenter specialties, and training levels. Such efforts could identify training strategies or team cultures and treatment philosophies that could be leveraged, as well as specific areas of weakness to be addressed. Additionally, ongoing research is needed to investigate when and how to consider chronological age at MCCs. As mentioned above, previous studies have identified an association between advanced age and worse treatment outcomes, yet a small but growing body of literature has also raised a concern that age-focused discussions may increase the risk of ageist decision making. Finally, another area to explore would be the ultimate impact of frailty- and geriatric-assessment-informed discussions on management decisions in geriatric oncology; this research could help reinforce the significance of purposeful language use in the collaborative care of older adults with cancer. 

## 5. Conclusions

This study found that MCCs often lack standardized language or objective assessments when describing the fitness and frailty of older adults with cancer. With an over-reliance on chronological age and comorbidity status to characterize a patient’s projected treatment tolerance or their overall health, practitioners attending MCCs may be at risk of age-based decision making. By incorporating frailty measurements from validated tools and/or by exploring multiple GA domains during case presentations, MCCs may be able to foster more comprehensive discussions that reflect important geriatric considerations and reduce ageist biases. Future work on the downstream effects of frailty- and GA-informed MCC presentations may further elucidate the role language can play in impacting clinical decisions and outcomes.

## Figures and Tables

**Figure 1 cancers-16-01477-f001:**
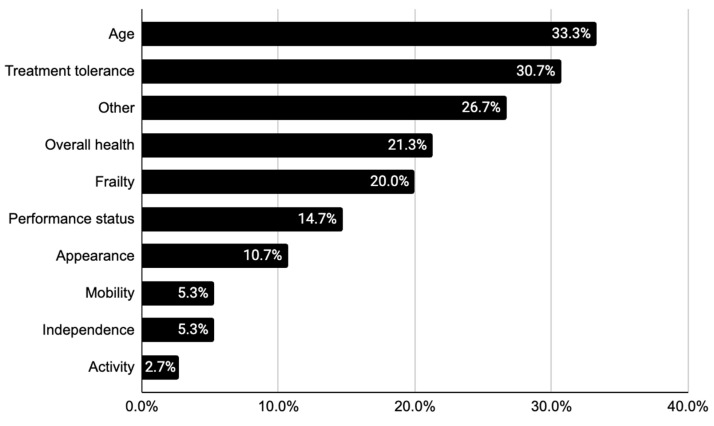
Descriptions of and synonyms for frailty across all case presentations (n = 75).

**Figure 2 cancers-16-01477-f002:**
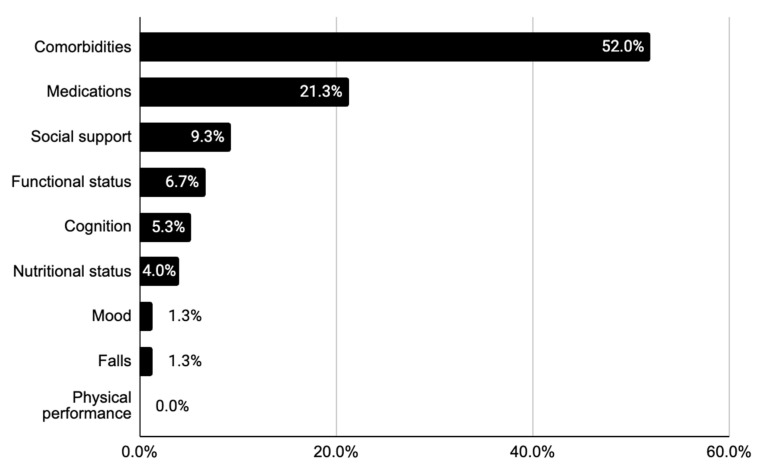
Mentions of GA domains across all case presentations (n = 75).

**Table 1 cancers-16-01477-t001:** Definitions and examples of frailty-related and GA-related variables.

	Variables	Definition	Examples
Frailty-Related	Frailty status	Description of patients as “frail”, “vulnerable”, or “fit”	“Quite frail”. “Pretty fit”.
	Age	Reiteration of age to suggest or emphasize frailty/fitness status	“Fine for age”. “He’s 86, but an okay 86”.
	Activity level	Description of patients as “active” or “inactive”	“Pretty healthy, very active”.
	Appearance	Description of patients’ appearance	“Looks younger than 80”. “Does not look like a candidate for anything”.
	Independence	Description of patients as “independent” or “dependent” on others, without elaboration on specific ADLs or IADLs	“Fully independent”.
	Mobility	Description of patients’ ability to ambulate or move	“Very vigorous, moves well”.
	Overall health	Description of patients’ general health or well-being	“Otherwise quite healthy”.
	Performance status	Mention of patients’ performance status per ECOG	“ECOG 1”. “Performance status poor, very symptomatic”.
	Treatment tolerance	Prediction of a patient’s ability to tolerate treatment	“No way fit for [chemotherapy]”.
GA-Related	Comorbidities	Mention of patients’ comorbidities (complete or incomplete) or comorbidity burden	“Many comorbidities, not a surgical candidate”.
	Medications	Mention of patients’ medication list (complete or incomplete)	“On apixaban”.
	Cognition	Mention of the presence/absence of any cognitive impairment	“Baseline dementia”.
	Falls	Mention of patients’ falls or fall risk	“Had a fall and [subsequent] hip fracture with surgical repair”.
	Functional status	Description of patients’ ability to perform activities necessary for or desirable in daily life	“Fully independent, drives here for clinic appointments”.
	Mood	Mention of patients’ mood	“[Patient] has generalized low mood”.
	Physical performance	Mention of objective measures of patients’ physical function (for example, gait speed, grip strength)	N/A
	Nutrition	Mention of patients’ appetite or nutritional status	“Struggling with malnutrition”.
	Social support	Mention of a social support network, including mention of caregivers or loved ones involved in the patient’s life	“Difficult social situation, [patient] lives alone, [with] family in [another country]”.

Note: “ADL” = activities of daily living, “IADL” = instrumental activities of daily living, “ECOG” = Eastern Cooperative Oncology Group, “GA” = geriatric assessment, “N/A” = not available.

**Table 2 cancers-16-01477-t002:** Demographic and clinical characteristics of older patients discussed at MCCs (n = 75).

Mean patient age, years (range)	80.9 (75–97)	
Patient sex, n (%)	Male	46 (61.3)
	Female	29 (38.7)
Presenter specialty, n	Medical oncology	22
	Radiation oncology	11
	Surgical oncology	45
Presenter level, n	Trainee	31
	Staff	38
	Both	2
Case presentations by site, n (%)	General gastrointestinal	4 (5.3)
	Genitourinary	19(25.3)
	Head and neck	19 (25.3)
	Thoracic	15 (20.0)
	Upper gastrointestinal	18 (24.0)
Presentation question, n (%)	Diagnostic	4 (5.3)
	Therapeutic	65 (86.7)
	Both	6 (8.0)
Cancer diagnosis, n (%)	Symptomatic	40 (53.3)
	Incidental	3 (4.0)
	Detection via surveillance	17 (22.7)
	Unclear or unspecified	15 (20.0)
Cancer treatment history, n (%)	New diagnosis	36 (48.0)
	Prior treatment	37 (49.3)
	Unclear or unspecified	2 (2.7)
Proposed treatment intent, n (%)	Curative	1 (1.3)
	Palliative	12 (16.0)
	Both	6 (8.0)
	Unclear or unspecified	56 (74.7)
Patient preferences, n (%)	Mentioned	15 (20.0)
Proposed treatment modalities, n (%)	Chemotherapy	35 (46.7)
	Radiotherapy	40 (53.3)
	Surgery	37 (49.3)
	Immunotherapy	4 (5.3)
	Best supportive care	6 (8.0)

**Table 3 cancers-16-01477-t003:** Presentation of frailty-related descriptors and GA domains in overall cohort (n = 75).

Frailty-related descriptors	Frailty status, n (%)	15 (20.0)
	Age, n (%)	24 (32.0)
	Activity level, n (%)	2 (2.7)
	Appearance, n (%)	8 (10.6)
	Appearance, n (%)	4 (5.3)
	Independence, n (%)	4 (5.3)
	Mobility, n (%)	16 (21.3)
	Overall health, n (%)	12 (16.0)
	Performance status, n (%)	15 (20.0)
	Treatment tolerance, n (%)	23 (30.7)
GA domains	Comorbidities, n (%)	39 (52.0)
	Medications, n (%)	16 (21.3)
	Cognition, n (%)	4 (5.3)
	Falls, n (%)	1 (1.3)
	Functional status, n (%)	5 (6.7)
	Mood, n (%)	1 (1.3)
	Physical performance, n (%)	0 (0)
	Nutrition, n (%)	3 (4.0)
	Social support, n (%)	7 (9.3)

## Data Availability

Restrictions apply to the availability of these data. Data were obtained from University Health Network and are available with permission from the Research Ethics Board of University Health Network.

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
