# Peer review of "Exploring the Language Used to Describe Older Patients at Multidisciplinary Cancer Conferences"

_cancers, 2024, doi:10.3390/cancers16081477_

Round 1
Reviewer 1 Report
Comments and Suggestions for Authors
In this study researchers sought a better understanding of the language used to discuss the management of complexities experienced by older adults with cancer, at multidisciplinary cancer conferences (MCCs). In general, this is a significant area or research due to the impact that it may have on the treatment or lack thereof that an elderly patient may receive based on the perceptions of their care team. Researchers provide detailed evidence which suggest that MCC case presentations disproportionately rely on chronological age and comorbidity burden with a lack of standardized language to define and explore other geriatric considerations relevant to frailty. In addition, standardized language should be adopted to improve outcomes in elderly patients diagnosed with cancer.
Author Response
We appreciate your thoughtful summary and positive comments on our paper
Reviewer 2 Report
Comments and Suggestions for Authors
This research report made us realize that at MCC, presenters and observers may have been developing discussions based on images formed solely by the patient's age (biases based on their own past experiences). I believe that by using the GA domains, we may be able to clarify the background factors of the presented cases and promote discussion from the perspective of geriatric medicine. I hope that this report will serve as an impetus for the formulation of presentation guidelines at MCC, and that the influence of latent age bias among physicians will be reduced in determining treatment policies in the field of geriatric medicine.
Author Response
We appreciate your thoughtful comments on our paper.
Reviewer 3 Report
Comments and Suggestions for Authors
Dear Authors,
I found this research to be interesting and potentially valuable for both researchers and clinicians. However, I have some points that I would like the authors to clarify:
1. Could the authors specify who attended each meeting? Were they doctors (residents/staff), nurses, or other professionals?
2. How were the meetings conducted? For example, who led the discussions?
3. Did all sites follow a similar pattern in conducting their meetings?
4. Was there any influence on the attendees, especially when they were aware of being observed by the research team?
5. Regarding frailty, is there a clear definition and standardized terminology used in the study, or is it still not well established?
6. About the statement on line 383 about "the need to improve discussions about older patients beyond their chronological age", I wonder if ageism (particularly among younger care providers) or biases related to sex, gender, and race may also be pertinent issues to consider.
